# What Are the Drug-Related Problems Still Faced by Patients in Daily Life?—A Qualitative Analysis at the Pharmacy Counter

**DOI:** 10.3390/pharmacy11040124

**Published:** 2023-08-03

**Authors:** Susanne Kaae, Ulla Hedegaard, Armin Andersen, Ellen Van Loon, Stijn Crutzen, Katka Taxis, Ramune Jacobsen

**Affiliations:** 1Department of Pharmacy, Faculty of Health and Medical Sciences, University of Copenhagen, 1172 København, Denmark; armin.andersen@sund.ku.dk (A.A.); ramune.jacobsen@sund.ku.dk (R.J.); 2Department of Health Research, Faculty of Health, University of Southern Denmark, 5230 Odense, Denmark; ulhed@medicinraadet.dk; 3Unit of Pharmacotherapy, Epidemiology and Economy, Groningen Research Institute of Pharmacy, Faculty of Science and Engineering, University of Groningen, 9712 Groningen, The Netherlands; wevanloon@kadds.nl (E.V.L.); s.crutzen@lentis.nl (S.C.); k.taxis@rug.nl (K.T.)

**Keywords:** patient agent, drug-related problem, qualitative research, community pharmacy services

## Abstract

Background: Drug-related problems (DRPs) affect many patients. Many activities in general practice, hospitals, and community pharmacies have been initiated to tackle DRPs. However, recent studies exploring what DRP patients are still facing in their daily lives are scarce. Methods: Danish pharmacy staff registered DRPs in prescription encounters to understand what DRPs patients are still experiencing in daily life. They noted short descriptions of what happened in the encounter that qualified the incident as a DRP. The descriptions were subjected to an inductive content analysis. Results: A wide range of DRPs that impacted patients’ daily lives practically and healthwise were identified. In total, eighteen percent of patients with prescriptions had a DRP. Three overall stages of DRPs were identified: challenges in receiving the medications, not knowing how or why to take the medications, and not experiencing satisfactory effects. Patients were emotionally affected by these problems. Conclusions: DRPs are still widespread in patients’ daily lives and influence their well-being. The identified DRPs illustrated the complexity of obtaining medications to work as intended and demonstrate that health professionals must take even the basics of medication intake much more seriously.

## 1. Introduction

Medications are the most common commodity to treat or prevent health-challenging conditions. However, obtaining the right medical treatment and obtaining the desired effect without too many adverse events is not easy. The more medications a patient receives, the more difficult the task becomes [1]. A German study showed that approximately 84% of polypharmacy patients suffer from a drug-related problem (DRP), and DRPs are known to cause approximately one-tenth of all hospital admissions [2]. DRPs are thus affecting the lives of millions of patients worldwide and are extremely costly [1].

A DRP has been defined by the Pharmaceutical Care Network Europe (PCNE) as “an event or circumstance involving drug treatment that actually or potentially interferes with the patient experiencing an optimum outcome of medical care” [3]. The burden of DRPs is increasingly being recognized, and initiatives to tackle the problems are being launched. For example, in Denmark, new counseling services in community pharmacies, both for new medicine users and for patients with chronic conditions, have emerged [4], the numbers of clinical pharmacists in hospitals have increased [5], and general practitioners (GP) are remunerated for conducting yearly medication reviews for patients suffering from chronic diseases [6]. Hence, one could expect that DRPs are moving in a desirable direction. On the other hand, society in these years also faces an aging population, a rise in health inequalities, and problems maintaining the productivity of health services. The question, therefore, is “What are the DRPs that patients still struggle with on a daily basis?”.

To obtain an overview of DRPs experienced by patients, several DRP classification systems exist [7,8]. DRPs are often identified by conducting medication reviews. The reviews can take different forms according to the type of data material, and setting. The reviews thereby span from an “ad hoc” approach consisting of a few situation-specific questions to a patient, for example, at the pharmacy counter or during a GP consultation, to cooperation between the pharmacist and/or the GP and the patient to systematically review all of the patient’s medications [9].

Another relevant type of activity to gain knowledge about DRPs is qualitative research, for example, anthropological studies to understand how patients perceive living with medications. Medications’ profound impact on patients’ daily lives has been shown; it includes how the medications influence patients’ self-perception, dilemmas between living with medications as “required to” as opposed to living as “wanting to”, and how patients must create daily routines and often worry about their medications [10,11,12]. However, DRPs covering patients” own experiences with medications are rarely operationalized into the existing DRP classification systems and might thereby be overlooked. Gyllensten et al. (2021) concluded that “most documents about pharmaceutical care are centred on the outcome of safer medication use rather than the underpinning ethics and practices of person-centredness” [13]. Hence, to advance initiatives towards tackling DRPs, there is a need to obtain an overview of the current situation of DRPs in approaches that also express patients” own experiences of DRPs as compared to the predetermined classification systems.

A relevant place to collect data on DRPs to understand how patients struggle with them on a daily basis is in the pharmacy. Here, you can employ the “ad hoc” approach to openly capture DRPs as experienced by the patient in that situation and register DRPs across therapeutic areas. However, only a few studies during the last decade have identified DRPs at the pharmacy counter, and they used predefined DRP categories to register the data [14].

The aim of this study was therefore to obtain an overview of the DRPs that patients are experiencing in daily life by keeping an open approach to the problems.

## 2. Materials and Methods

This paper was written according to the principles of the “Standards for Reporting Qualitative Research” [15].

### 2.1. Design

Already in 1988, did Strand et al. advocate for the need to keep increasing our insights into DRPs by not only retrospectively classifying them but also registering DRPs concurrently with daily life and practice, i.e., documenting the DRPs as they are happening to learn from them [16]. Furthermore, van Mil et al. (2004) recommended, in recognition of the complexity of DRPs, that when registering DRPs, the option of free text should always be included [7]. To understand DRPs and what they cover more in line with how patients actually experience them, we therefore conducted a study in community pharmacies, i.e., based on the “ad hoc” approach, by staff noting down DRPs in their own words when they observed it at the counter, as recommended by Strand and van Mil, to better grasp the complexity of the DRP and the possible influence of it on the patient.

### 2.2. Setting

The study was conducted in Denmark in 11 community pharmacies. In total, there are 525 community pharmacies in Denmark. Each Danish community pharmacy serves more than 90,000 customers yearly and has an average of 29 employees [17]. Prescription medications in Denmark are divided into two categories: those reimbursable by the state and those not reimbursable. If reimbursable, the prices of the medications are totaled, and only when exceeding an out-of-pocket expense of 1020 DKK (137 EUR) yearly for adults is a reimbursement rate of 60% applied. The maximum out-of-pocket payment per year is 4320 DKK (581 EUR) [18]. The prescription system is electronic.

### 2.3. Data Collection Instrument

The registration of DRPs was made in a semiqualitative design, both quantitatively identifying the total DRPs as well as more openly describing the incident, as described above, by the use of free text. The definition of a DRP by PCNE was used in the project. Only DRPs discussed directly with the patient in a face-to-face encounter in the pharmacy were registered in the study, and only prescription medication was included. The DRP was registered independently of whether the patient or the pharmacy staff member started the conversation about the problem.

### 2.4. Data Collection Procedure

Pharmacy staff registered all of their prescription encounters, including those with DRPs, for a total of 6h. Staff had to register DRPs in a developed online system (SurveyXact). As the data collection was part of a bigger study exploring the effects of new postgraduate training for pharmacy staff, it was collected twice during a period of 5 months.

If a DRP was identified, oral consent to register different data was obtained from the patient before proceeding. For patients with positive consent, the following information was registered: sex, birth year, prescription status (first time or refill), name of the drug, a short description explaining the DRP, and any other information that the staff participant wanted to share with the researchers about the case. As the registration, as described above, was conducted as part of a pre/postintervention study of a new post-curriculum education in Denmark and the Netherlands, staff also registered the type of DRP according to eight main categories in the DOCUMENT system [19] and which solution staff had applied according to five categories. However, these data were not used for the purpose of this study.

### 2.5. Analysis

To understand what the registered DRPs covered, a qualitative analysis based on the notes that staff recorded at the counter was performed. We applied inductive content analysis to be open to the daily problems encountered with medications. NVivo + was used to organize and code the data [20]. The open coding implied trying to understand the problem around the medication, including any observed effect on the patient as revealed in the descriptions. Due to the type of data, i.e., the stories were short and could not be further explicated, the analysis focused on the manifest contents displayed in the data rather than interpreting them [21].

The analysis was performed in several steps. As part of the initial coding of the small stories of identified DRPs, notes and reflections on the understanding of the codes were made. After the first set of coding, a refining process involving the notes was conducted by reorganizing and clustering all codes. One researcher developed a coding scheme for the main categories, generic categories, and subcategories (SK). These results were then confirmed by performing triangulation. Hence, RJ compared the developed coding scheme with raw data and assessed (a) whether most of the data could be coded according to the categories/whether the categories covered the data sufficiently and the abstracting/coding process was thereby complete [20] and (b) whether the final descriptions of the categories were justifiable by the data [21]. Based on this process, some data were recoded and reorganized.

## 3. Results

The data collection occurred in Denmark in September 2021 and in January 2022 in the 11 pharmacies, involving 28 pharmacy staff members. The staff members comprised 1 male and 27 female staff, spanning 14 pharmacy technicians, 13 pharmacists, and 1 pharmacy owner (also a pharmacist).

A total of 1806 prescription encounters were registered, of which 319 (18%) were recorded as DRPs. Twenty-two of the patients for whom staff registered a DRP did not give their consent to register any details; hence, the qualitative analysis was based on 297 cases.

Three overall stages of problems with medication use were identified (see Table 1 for an overview). First, problems were identified with the patient receiving the medication. Second, after the patients had received the medication, there was a problem with the patient missing basic information about how and why to take the medication. Third, when both the medication and basic information about the usage had been received, problems with obtaining satisfactory effects from the medication were registered. The described impacts on patients’ lives show how the medications affected patients not only physically, for example, by experiencing side effects, but also emotionally and practically. Below, the results are structured according to the three stages of medical problems and explained in greater depth with regard to the various aspects of the DRPs, followed by their influence on the patient’s life.

### 3.1. Problems with Receiving the Medication

A frequent problem for patients was not receiving the prescribed medication. Hence, this category covered 64 cases (22% of the registered DRPs). This type of problem pertains to patients not picking up the medication, the prescriber not ordering it, or problems with the distribution system. This affected patients in the practical aspects of handling medication but also disturbed them emotionally to some degree.

#### 3.1.1. Description of Problems with Receiving the Medication

Some patients had difficulties receiving refill prescriptions, i.e., remembering to ask their doctor to renew the prescription or to pick up the ordered medication in the pharmacy. Furthermore, some patients forgot to take medication after they had purchased it. A few patients stated that problems paying for the medication affected whether they picked up the medication and used it as often as intended.


*“The patient has forgotten to hand in the prescription for atorvastatin. Has therefore not taken the medicines for 2 days. Talked about dosage dispensing boxes” (refill prescription for atorvastatin)—DRP no. 1208*



*“Often forgets a tablet” (refill prescription for Mini-Pe [ed. Norethisterone])—DRP no.1287.*



*“Due to the price, the patient doesn’t use it every day” (refill prescription for Xarelto [ed. rivaroxaban)—DRP no. 167.*


There were, however, also many situations in which the patient did not receive the medication because it was unavailable when they arrived at the pharmacy. This was either caused by the doctor not issuing the prescription in the electronic system or delivery problems.


*“There was no prescription. Risk of undertreatment” (refill prescription for Ventoline [ed. salbutamol])—DRP no. 1509.*



*“Delivery problems—delivery date unknown” (refill prescription for Estrogel [ed. oestradiol])—DRP no. 1285.*


#### 3.1.2. Influence on the Patient’s Life Caused by Problems Receiving the Medication

Further problems with purchasing the medication and thus receiving it were situations in which the prescription contained other types of medications than the patient expected. This resulted in some patients being unsure and confused about the right medication to receive.


*“The patient went to collect medicine for high blood pressure. The patient was treated with three different products. Prescriptions are missing for ramipril 10 mg and metoprolol succinate 100 mg. The patient said that the dosage of amlodipine had been increased from 5 mg to 10 mg. Unfortunately, this is not stated in the system. The patient is very insecure about blood pressure treatment. He believes that he uses a diuretic. That is not stated in the system. Have recommended that he speaks with the doctor and get dosage-dispensing services and gave him a brochure about it” (refill prescriptions for amlodipine, ramipril, and metoprolol succinate)—DRP no. 1040.*


Hence, patients often went to the pharmacy in vain because there was no prescription, or it was unclear what medications they should receive; therefore, they or the pharmacy staff had to spend additional time calling the prescriber. In addition, some negative feelings were registered concerning receiving many medicines, both in terms of the practicalities of organizing them and how patients felt about taking the medication. However, patients who were emotionally affected by forgetting to take medication were not described.


*“Thinks he has to take many pills every day, and it’s a hassle to keep track of it” (refill prescriptions for Velmetia [ed. metformin; sitagliptin], Jardiance [ed. empagliflozin], and several blood pressure products)—DRP no. 798.*



*“The dosage is two tablets four times a day. She thinks it’s a lot of tablets every day” (refill prescription for quetiapine)—DRP no. 862.*


### 3.2. Problems with Knowing How to Take the Medications

When receiving the medication, a considerable number of registered DRPs expressed a lack of clarity about how to take it. This category was, in fact, the most frequent type of problem, covering 176 cases (59%) of the registered DRPs. The category included several cases in which the patient, according to the staff, took the medications incorrectly. This type of DRP affects patients emotionally and physically.

#### 3.2.1. Description of Problems: Not Knowing How to Take the Medications

There was a widespread lack of fundamental knowledge from the patients about how many prescription medications to take, how often to take them, and for how long. There were several cases where the patient did not even know why he or she had been prescribed the medication. Lack of basic information pertained both to starting a new single-drug treatment and refilling prescriptions of different types of medications for the same chronic disease.


*“He didn’t know how to take it” (first-time prescription for paracetamol)—DRP no. 1545.*



*“Didn’t understand why he should take it. What disease did he have” (first-time prescription for Eltroxin [ed. levothyroxin])—DRP no. 1440.*



*“Patient has picked up the Kaleorid [ed. potassium chloride] 9 days ago together with Furix [ed. furosemide]. The patient has taken Furix as written on the prescription but has not taken Kaleorid because he didn’t know what it was and why to take it” (refill prescription for Kaleorid)—DRP. No.171.*


Staff further registered that patients, apart from lacking the basic knowledge of how to take medication, often also needed additional information about technical aspects of the treatment, such as whether the medication should be taken with/without food or how to store it. Furthermore, there were examples where staff provided information broader than discussing basic information about intake and usage. This was advice related to self-care where broader aspects of the disease, not just the medication, were brought into the discussion, sometimes by request from the patient. Hence, staff noticed that some patients were unaware of these issues.


*“The patient didn’t know how to use it and how to store it (in the fridge)” (first-time prescription for Brentacort [ed. hydrocortisone; miconazole])—DRP no. 442.*



*“The patient doesn’t know how to apply it to the eye. She knows it has to be used four times daily but doesn’t know how to “put in in’ the eye” (first-time prescription for Kloramfenikol [ed. chloramphenicol])–DRP no. 594.*



*“The patient had received Penomax [ed. pivmecillinam] before but did not know the most important advice for self-care. Was very happy about the information” (refill prescription for Penomax)—DRP no. 1669.*


Another type of lacking information was in connection with refill prescriptions. Hence, over time and sometimes across different providers, misalignment in the understanding of how to take medication developed, particularly about the correct dosages, leading to confusion.


*“On the box, it says one tablet daily, but the new one from the doctor says two tablets daily. She thinks she and the doctor previously did discuss a change but is not sure” (refill prescription for losartan)—DRP no. 1077.*



*“As it was an old prescription, there was some confusion around the dosage, but we had it sorted out. She just had dosage changes, so it was different from the prescription” (refill prescription for Eltroxin [ed. levothyroxine])—DRP no. 100.*


Other problems regarding taking medication correctly were situations in which staff noticed that the patient took medication incorrectly, for example, that the patient took too little of the medication, incorrectly applied the medication, or did not take recommended supplementary products. Hence, these were situations in which the patients already took the medication (or, if for first-time use, had made up their mind about how to use it) and believed that their intake was appropriate.


*“Comes from the GP and was just informed that it’s twice daily, so has the intention to stop when it’s gone. Therefore, I advised continuing until 10 days after every visible sign had disappeared” (first-time prescription for Brentan [ed. miconazole])—DRP no. 69.*



*“The patient doesn’t think the drug has any effect. The prescription says one tablet two times per day. The patient only takes one tablet in the morning, together with breakfast. The tablet should be taken on an empty stomach to have the optimal effect” (refill prescription for pantoprazole)—DRP no. 265.*


Situations were described in which discrepancies between the staff’s and the patient’s understanding of how to take the medication occurred, mostly with regard to refill prescriptions. In some cases, the patient admitted to having misunderstood elements of the treatment and expressed gratitude towards its detection, and in others, although in fewer cases, patients openly objected to the staff’s advice. Staff also noticed incorrect medication intake due to errors made directly by the prescriber, such as prescribing wrong dosages, inappropriate drug combinations, or, on a more practical note, prescribing tablets that the patient could not swallow.


*“The patient hands in a prescription on both Xyzal [ed. levocetirizine] and cetirizine. I am surprised he has been prescribed to use both products at the same time” (first-time prescriptions for Xyzal and cetirizine)—DRP no. 472.*



*“Problems swallowing it—is not coated” (refill prescription for amlodipine)—DRP no. 1225.*


#### 3.2.2. Influence on the Patient’s Life Caused by Not Knowing how to take the Medications

The situation of lacking information about correct medication intake often affected patients. Sometimes, it was stated factually and neutrally that the patient did not know how to use the medications. However, in other cases, the lack of information about how to take the drug, resulting in the patient being emotionally affected, was more explicitly described. “Having doubts about”, “being insecure about”, and “being confused” were frequently reported feelings in the encounters.


*“Hadn’t been informed about the dosage” (first-time prescription for Priminova [ed. phenoxymethylpenicillin])—DRP no. 246.*



*“The patient is unsure how long to drip in the eye” (first-time prescription for Spersadex Comp [ed. chloramphenicol; dexamethasone])—DRP no. 575.*



*“The patient seemed confused and didn’t know what to do with the new tablets, so she asked for advice” (first-time prescription for valaciclovir)—DRP no. 663.*


In some cases, not knowing how to take the medications led to patients experiencing negative physical consequences.


*“It says one tablet daily in the prescription. I ask him when he takes it during the day. In the evening, he answers. I ask if he has to get up to pee during the night. “Yes, has it something to do with that?” he asks” (refill prescription for Centyl [ed. bendroflumethiazide])—DRP no. 94.*


### 3.3. Problems with Obtaining Satisfactory Effects

A third frequent type of DRP detected was when patients experienced some problems after receiving and taking the medication. Hence, this category involved 84 cases (28% of the identified DRPs). The most registered examples were patients who experienced a lack of effect or side effects. Hence, these types of problems were related to symptoms experienced by the patient during the treatment. Staff also noticed how some patients were often left unattended while pursuing good medical treatment. This type of DRP affected patients physically and, to a high degree, emotionally.

#### 3.3.1. Description of Problems Obtaining Satisfactory Effects

The experience of a lack of anticipated effects with the medication was common. This led to patients not being relieved of the usual troubling physical symptoms.


*“He doesn’t think it works so well any longer. The pain has gotten a bit worse” (refill prescription for gabapentin)—DRP no. 1342.*



*“The patient doesn’t feel pain-relieved” (refill prescription for Buprenorfin [ed. buprenorphine])—DRP no. 1108.*


Another frequent type of negative experience with medication was side effects. Some cases described how patients were bothered by the side effects of daily life physically and, in some cases, also mentally. Patients were often unaware that the symptoms they suffered from were due to the medications.


*“The patient experiences side effects from the stomach and intestines; she thinks it’s bothersome” (refill prescription for metformin)—DRP no. 339.*



*“Asked if he experiences problems from the mouth. Drinks 10 litres per day. “No one ever told me about this before”. He does not see a dentist regularly. Advised about saliva function. We looked together at the possibilities from the pharmacy, and he got a sample of saliva-inducing tablets” (refill prescriptions for Elvanse [ed. lisdexamphetamine] and quetiapine)—DRP no. 941.*


Patients’ problems with side effects also pertained to situations that concerned patients’ suspicion of experiencing them in the future. The anticipation of a side effect that the patient had not experienced earlier could be as strong as the anticipation of a side effect experienced in a previous treatment.

#### 3.3.2. Influence on the Patient’s Life Caused by Problems Obtaining Satisfactory Results

The feelings attached to the anticipation of experiencing side effects were the strongest of the many registered feelings across the data. Feelings such as “fears”, “worries”, and “strong nervousness” were described in these cases.


*“First time—afraid of getting nausea” (first-time prescription for Brintellix [ed. vortioxetine])—DRP no. 1216.*



*“The patient cries and is nervous to start because she is afraid of side effects” (first-time prescription for methylphenidate)—DRP no. 509.*



*“The patient is very nervous about continuing to take them—they are very expensive, and he experiences get many blue marks on hands and arms—is afraid that he will get bleedings in the brain” (refill prescription for Eliquis [ed. Apixaban])—DRP no. 106.*


One problem identified in ensuring the good effects of the medication and dealing with the negative experiences of it was that staff, in several instances, noticed that the patient had not been summoned for a follow-up. Hence, patients were left unattended with their medications and their worries. This sometimes meant patients received medications that interfered with their daily lives, as shown above, which only pharmacy staff noticed.


*“Hadn’t been told to control the blood sugar, as it is a new treatment” (first-time prescription for Ozempic [ed. semaglutide])—DRP no.102.*



*“The patient hasn’t been to any kind of control at the doctor since she started the asthma medication close to 2 years ago” (refill prescriptions for Spirocort [ed. budesonide] and Bricanyl [ed. terbutaline])—DRP no. 1254.*



*“The patient works as a driver and has to take oxycodone for one month due to pain. Red triangle. Hasn’t tried other types of analgesics for his knee” (first-time prescription for oxycodone)—DRP no. 1385.*


Medications sometimes impacted patients’ lives so much that they wanted to stop taking them. In addition to the fear of side effects, other reasons for not wanting to take the medication were that the patient thought the medication was no longer necessary, a lack of trust in the effects of the medication, or simply a dislike of medication.


*“Resistance from the patient in starting up; earlier had received simvastatin” (first-time prescription for rosuvastatin)—DRP no. 1165*



*“The dermatologist has prescribed Dermovat [ed. clobetasolpropionate] (4 g) cream for itching eczema (cause unknown). Earlier received Locoid [ed. hydrocortisone-17-butyra] (2 g), Elocon [ed. mometasonfuroat] (3 g) and systematic prednisolone from our own doctor without any effect. Is summoned to control in 3 months. She appeared to have given up and had no faith in the new cream. Will now supply it with a fatty cream. Advised to contact the doctor before time, if there is no effect” (first-time prescription for Dermovat)—DRP no. 938.*



*“The patient has taken this medicine since her husband died. Now she wants to stop, but the doctor recommends waiting for the spring. The patient is very disappointed and would like to stop now. I fear because of the antipathy against the medicine that she stops the treatment” (refill prescription for Paroxecare [ed. paroxetine])—DRP no. 649.*


## 4. Discussion

The results show that DRPs were widespread, as nearly one out of five patients with prescriptions experienced them. Fundamental challenges in receiving the medications and not knowing how to take them or why were common. Furthermore, obtaining satisfactory effects from the medication was a challenge. The influence of these problems on the patient was at a practical, emotional, and/or physical level. These different types of DRPs illustrated the complexity of obtaining sometimes even a single-drug treatment to function as intended and the considerable efforts needed from the patient, the GP, and pharmacy staff to obtain the right treatment.

### 4.1. The Medication Problems Experienced by Patients in Everyday Life

This study identified a number of problems also reported in other studies investigating DRPs using predefined classification tools, such as patients’ knowledge gaps about the use of medications, the uncertainty of patients regarding the aim of their medications [2,22], the unavailability of prescribed medication [14], and the experienced side effects [23]. However, where most classification systems capture just one or two of these aspects, the open design of this study allowed us to capture them all and organize them into a new three-stage model. Hence, this study demonstrates how all of these problems exist on a daily basis and how they sometimes constitute a continuum of interrelated problems.

Furthermore, some problems, typically reported in qualitative studies, appeared. One such problem pertained to organizing medications to receive them. Swinglehurst and Fudge (2021) pinpointed the often hidden need of patients to develop strategies to manage their medication, particularly when many types of medication are involved [10,24], and recommended “greater appreciation among prescribers of the nature and complexity of this work” [10]. Another such result that was also captured in our study, but otherwise usually only uncovered in in-depth qualitative studies, was how negative emotions are involved in taking medication. In particular, we show how patients become emotionally affected by not understanding how or why to take medications and how the fear of potential adverse events is especially strong.

On the other hand, nonadherence is a DRP frequently reported in the scientific literature [25], but this problem was not a prominent one in this study. This difference might have to do with the fact that the data in this study were collected at the time of dispensing. However, the difference might also stem from the fact that nonadherence is more of a concern for health care professionals than patients. Ryan et al. described how “to integrate a regulation (such as adherence to medication), people must grasp its meaning and synthesize that meaning with respect to their other goals and values” [26]. Thus, many patients may not see nonadherence as a problem because they do not see a strong value in receiving and continuing their medications [12,27]. This point is supported by the number of patients in our study not knowing why and how to take their medications; hence, prescribers and pharmacy staff have failed to fully explain to people why and how to take their medications in a way that also resonates with the values of the patient. Hence, a reason why we did not find nonadherence as a problem in this study was perhaps that the results were indeed, and as intended by the open design, more in line with how patients see their own medication.

### 4.2. Limitations and Strengths

The results of this study were based on a large amount of real-life data, i.e., data collected concurrently with daily practice across therapeutic areas and types of prescriptions, thereby increasing the “credibility” of the data. Furthermore, we installed processes to ensure the “confirmability” of data by introducing researcher triangulation. We have likewise tried to be transparent about the methodological procedures and decisions made during the project to account for the “dependability” [28].

The data were not very rich and thereby left out the possibility of conducting analyses beyond the manifest level. This aspect limits thick descriptions and thus insights into the potential “transferability” of the data [28].

The simple data collection instrument with staff writing down identified DRPs in their own words was a highly efficient method to both identify many different types of DRPs across a number of existing DRP classification tools and uncover problems, possibly more in line with patients’ own experiences. The open “ad hoc” data collection approach might therefore be considered for more frequent use in the future.

Some further implications could now be to employ more projects investigating how to address patients’ emotions in the pharmacy, as they play a vital role in taking medications. Practical implications also include the necessity for pharmacies to reconsider all types of DRPs that patients are experiencing in daily life, even the most fundamental ones, and how they can be recognized and handled. In this process, staff should not assume that even the most basic information about drug usage, including the aim of the treatment, has been adequately communicated to the patient and should therefore prioritize helping the patient better understand why and how to take their medications.

## 5. Conclusions

DRPs are widespread in patients’ daily lives and influence their well-being. Challenges in receiving the medications, not knowing how and why to take them, and obtaining satisfactory medical effects were identified. The identified DRPs illustrate the complexity of obtaining medications to work as intended but also suggest that health care professionals must take the fundamentals of medicine much more seriously. Pharmacy staff and prescribers have an important task in ensuring these aspects.

## Figures and Tables

**Table 1 pharmacy-11-00124-t001:** Stage of DRP and influence on the patient’s life.

**Stages of the DRPs**
Problems in the following:
1. Receiving medication
2. Knowing how to take the medications
3. Experiencing satisfactory effects from the medication
**Influence on the Patient’s Life**
*Practically affected*
Need to keep track of medication ordering and intake
Use of additional time to call the prescriber and revisit the pharmacy
*Emotionally affected*
Uncertainty and confusion about how to take the medications
Negative feelings of lack of trust, fears, and worries about the medications
*Physically affected*
Persistent symptoms due to lack of effects
Symptoms due to side effects

## Data Availability

Data is available upon request to the main author.

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
