# Peer review of "What Are the Drug-Related Problems Still Faced by Patients in Daily Life?—A Qualitative Analysis at the Pharmacy Counter"

_pharmacy, 2023, doi:10.3390/pharmacy11040124_

Round 1

Reviewer 1 Report

Dear Authors please see my comments below:

1) Why the data collection had a long duration? please clarify the reasons within the manuscript. 

2) The data collection took place back in 2021, please clarify why the manuscript is submitted now in 2023 and not earlier.

3) The feedback/comments need a better organisation, probably all summarized in a big table

4) Conclusion: clarify why is better for the staff to write on their own words, it needs a brief description.

Author Response

We thank the reviewer for his/her valuable comments which we tried our best to accomodate. Please see below.

  • Why the data collection had a long duration? please clarify the reasons within the manuscript. 

The data was collected a part of another study which is now explicated in the method section, which therefore accounts for the data collection period.

  • The data collection took place back in 2021, please clarify why the manuscript is submitted now in 2023 and not earlier.

We tried to submit the manuscript first to another journal where is after a longer review period was rejected, that’s way the submission only took place in the first half of 2023.

  • The feedback/comments need a better organisation, probably all summarized in a big table

To help the reader better overview and understand the comments, we have now further divided the result section into the headings from the result table – hence we have sub-divided each of the 3 parts into further 2 parts: Description of the specific type of DRP; followed by: Influence in the patient’s life caused by the specific DRP. Further, we have re-arranged results slightly.

  • Conclusion: clarify why is better for the staff to write on their own words, it needs a brief description.

We have now tried to explain this aspect a bit further in the method section.

Reviewer 2 Report

The only difficulty I had in thinking about this paper was in relation to the difficulties sometimes encountered in the pharmacist receiving prescriptions, and I wondered if this might be a result of some sort of an electronic information transfer system which was in use in Denmark but perhaps not be generally employed in some other countries. I wondered if a short paragraph explaining the stages between the issuing of the prescription and its receipt by the pharmacist might help the reader in countries where the usual course is for a hard copy of the prescription to be received by the pharmacist prior to dispensing.

I wondered whether the pharmacists concerned raised the question of difficulty in relation to the prescription contents with the patients, or whether patients themselves initiated the matter.

It seemed to me that the first two of the three categories of the difficulties encountered largely related to the first dispensing of a particular drug, and the final category mainly to the subsequent dispensing. I think the information in relation to 1st dispensing had been recorded in your study and wondered if you looked further at the matter.

Author Response

We thank the reviewer for his/ hers valuable comments which we have tried our best to accomodate:

The only difficulty I had in thinking about this paper was in relation to the difficulties sometimes encountered in the pharmacist receiving prescriptions, and I wondered if this might be a result of some sort of an electronic information transfer system which was in use in Denmark but perhaps not be generally employed in some other countries. I wondered if a short paragraph explaining the stages between the issuing of the prescription and its receipt by the pharmacist might help the reader in countries where the usual course is for a hard copy of the prescription to be received by the pharmacist prior to dispensing.

Indeed, we have therefore added this info the description about the Danish pharmacy system.

I wondered whether the pharmacists concerned raised the question of difficulty in relation to the prescription contents with the patients, or whether patients themselves initiated the matter.

Both parties could initiate a conservation of the problem which we have now added as an info to the method section.

It seemed to me that the first two of the three categories of the difficulties encountered largely related to the first dispensing of a particular drug, and the final category mainly to the subsequent dispensing. I think the information in relation to 1st dispensing had been recorded in your study and wondered if you looked further at the matter.

Thank you for this comment, we did unfortunately not make an explicit data analysis into any detectable difference between 1st and refill prescription DRPs, though it could be very interesting.
